# Reformulation of Pastry Products to Improve Effects on Health

**DOI:** 10.3390/nu12061709

**Published:** 2020-06-07

**Authors:** Ramon Estruch, Eulàlia Vendrell, Ana María Ruiz-León, Rosa Casas, Sara Castro-Barquero, Xavier Alvarez

**Affiliations:** 1Department of Internal Medicine, Hospital Clinic, Institut d’Investigació Biomèdica August Pi i Sunyer (IDIBAPS), University of Barcelona, Villarroel 170, 08036 Barcelona, Spain; amruiz@clinic.cat (A.M.R.-L.); rcasas1@clinic.cat (R.C.); sacastro@clinic.cat (S.C.-B.); 2CIBER 06/03: Fisiopatología de la Obesidad y la Nutrición, Instituto de Salud Carlos III, 28029 Madrid, Spain; 3DALLANT, SA, Carrer de Laureà Miró 392, 08980 Sant Feliu de Llobregat, Barcelona, Spain; evendrell@dallant.com (E.V.); xalvarez@dallant.com (X.A.)

**Keywords:** pastry, bakery, cardiovascular health, hypertension, dyslipidemia, type 2 diabetes, sugar, trans-fats, salt

## Abstract

Obesity is increasing at an alarming rate and has been described as a global pandemic. This increase has several explanations, including an increase in caloric intake, low levels of physical activity and the nutritional composition of our diets. In addition to public health policies based on healthy dietary patterns and recommendations based on the Mediterranean and other healthy diets, food reformulation, especially of commonly consumed processed foods, such as bakery products and pastries, is needed in the fight against obesity. Among nutritional reformulation strategies, reductions in caloric density, salt, added sugar, saturated and trans-fats are important in order to reduce the associated risk of developing chronic diseases, including cardiovascular diseases, diabetes and cancer.

## 1. Introduction

Obesity is a serious public health concern that has reached epidemic proportions in most or almost all regions of the world [1]. The incidence of obesity [Body Mass Index (BMI) ≥ 30 kg/m^2^] worldwide has increased from 28.8% in 1980 95% Confidence Interval (CI: 28.4–29.3) to 36.9% in 2013 (CI: 36.3–37.4) for men and from 29.8% (CI: 29.3–30.2) to 38.0% (CI: 37.5–38.5), respectively, for women. In fact, the rising prevalence of overweight and obesity in several countries has been described as a global pandemic [1]. Worryingly, the prevalence of overweight and obesity in children has also increased in the last 10 years (from 2006 to 2016), especially in Southern European countries like Italy (32.4%; CI: 23.8–42.4), Greece (29.6%; CI: 14.5–45%) and Portugal (26.4%; CI: 23.8–29.2) [2]. In Spain, the European country with the lowest incidence of cardiovascular disease (CVD) [3], the increase has also been very large (22.6%; CI: 18.7–27.0). Thus, further significant increases in the prevalence of obesity and other cardiovascular risk factors can be expected in the following decades if we are not able to reverse these trends [4]. Potential contributors to the large increase in obesity over the past 30 years include increased caloric intake, changes in diet composition, declining levels of physical activity and changes in other factors, such as the gut microbiome [2]. In order to reduce morbidity, mortality and the expected costs of healthcare for obese adults, the main targets for the prevention and treatment of overweight and obesity should be children and young adults.

Over the past years, healthy diets such as the Mediterranean (MD), Nordic, Dietary Approaches to Stop Hypertension (DASH), and vegetarian diets [5,6], which are characterized by high food diversity and their nutritional profiles, have been dramatically displaced by unhealthy diets based on high-caloric foods and nutritionally poor profiles [7]. While healthy dietary patterns rich in fresh fruits, vegetables, legumes, pulses, and nuts have demonstrated the positive effects of diet in the prevention of the most prevalent chronic diseases, including CVD, diabetes mellitus, cognitive decline, dementia and even cancer [8,9,10,11]. Unhealthy diets are characterized by high intakes in animal products and processed foods, salt, simple sugars and saturated and trans fatty acids, and are directly associated with higher risks of non-communicable diseases (NCDs) and the onset of malnutrition [12]. The increase in unhealthy diets is widely attributed to these diets being cheaper and based on more widely available food items than healthier diets. Thus, in the last decades, a progressive decrease in adherence to the traditional MD has been observed in almost all Mediterranean countries, including Spain [13]. Therefore, governments, official institutions and scientific societies should continue promoting healthy dietary patterns together with strategies for restricting the rest of the dietary risk factors associated with obesity, CVD, diabetes and cancer, such as the intake of excess energy, salt, saturated and trans-fats, and sugar [14,15,16,17].

In the last years, the selling and distribution of food appear to be associated with the risk of developing NCDs, but their effects remain unclear. In a recent study carried out by Santulli et al. [18], it was observed that the length of the food supply chain might have a determining role in the prevention of metabolic syndrome (MetS). The authors reported that higher adherence to the MD, together with shorter supply chains, significantly reduced the prevalence of MetS.

On the other hand, unfortunately, according to official data from the Report on Food Consumption, in 2018, the purchase of bakery and pastry products by Spanish households increased by 0.6% compared to 2017. The main food consumed away from home when sorted by the number of servings was bread (with its intake rising to 30.6%), which was then followed by meat (24.0%), vegetables (23.8%) and pastries (15.0%). Thus, each inhabitant in Spain consumed 5.89 kilos of pastries and cakes at home in a year, representing an increase of 0.3% over the previous year. It is noteworthy that most of the consumption per capita came from the packaged segment, although this rate fell by 1.2% in 2017 [19]. In addition, World Health Organization (WHO) reports on food and beverage categories, which are based on food consumption surveys undertaken in the period 1990–2010 in Europe (49 countries were analyzed), showed significantly high intakes of cakes, sweets, biscuits and pastries (29%, Figure 1) [20]. According to this information, cakes and pastries are the most popular products offered by the pastry and bakery industries, and a global growth of around 2.5% is expected in this market in the period 2019–2025 (from USD 6.69 billion in 2018 to USD 8.15 billion by the end of 2025) [21].

Consequently, in 2004, the WHO recommended that the industry reduce their products’ energy densities, trans-fats, saturated fats, sodium and added sugars [22]. Although little progress has been made in this respect during the last decade, in 2014, the Dutch government asked the food sector to decrease the levels of energy, salt, saturated fat and sugar in food products (National Agreement to Improve Product Composition: Salt, Saturated Fat, Sugar Calories) [23]. Moreover, in 2018, AESAN (Spanish Agency for Food Safety and Nutrition) also recommended that the industry reformulate their products in the same manner and requested a 5% reduction in added sugar and saturated fatty acids in bakery products and pastries by 2020 [24]. The main aim of food reformulation is to develop healthier foods, thus benefitting human health. Reducing ingredients that are perceived of as being “harmful” to human health provides the chance to substitute “healthier” ingredients. Thus, food technology science drives effective food reformulation strategies for reducing the incidences of several chronic diseases associated with unhealthy dietary habits and improves food products’ nutritional profiles. Others aspects included in reformulation are increasing the content of health-promoting ingredients, such as whole grain flour, dietary fibre, fruits, vegetables and unsaturated fatty acids, that increase the nutritional quality of foods by improving their macro- and micro-nutrient compositions in terms of, for example, vitamins, minerals and phytochemicals [25]. Reformulation will only be successful if reformulated foods fit into a healthy diet and have high nutritional quality, are of good texture, safe, palatable and low in price [26]. Few studies have analysed the technological aspects of sodium, sugar and saturated fat reduction. This review analyses the possible effects of the reformulation of pastry products on health according to different reformulation targets.

## 2. Reduction in Salt

Manufactured food is the main dietary source of salt (75–80%) in industrial countries [27]. Salt (sodium) consumption is one the main determinants of all-cause mortality and cardiovascular disease [28,29]. High salt intake is associated with arterial hypertension, and a moderate reduction in salt consumption leads to a significant reduction in blood pressure (BP). On the other hand, well-conducted cohort studies and a few intervention studies have concluded that low salt consumption is associated with reduced cardiovascular outcomes [30]. In this context, the WHO guidelines recommend a 30% reduction in salt (Figure 2) consumption by 2025 [31], including an ideal target of no more than 5 g of salt per day (or a sodium intake of < 2 g/day). However, since low salt (sodium) intake has been associated with an increased risk of death due to cardiac issues, the sodium target levels should be higher than 1.2 g/day [29]. In fact, a number of prospective observational studies and two randomized clinical trials (ONTARGET and TRANSCEND studies) have suggested a J-shaped association between salt (sodium) consumption and the incidence of cardiovascular disease [32,33].

Sodium is one of the main contributors to the palatability of a product. Its use increases saltiness and improves the aroma and taste of several aromatic compounds; bitterness is mainly decreased by cross-modal interactions with sodium [27].

Meat products, bread and bakery products are the most common sources of salt intake, although other food groups, such as processed meats, dairy products and sauces, also contribute. Salt is crucial for taste and dough structure, but spices, potassium chloride, monosodium glutamate, nucleotides (inosinate and guanylate), other glutamate salts, umami taste, yeast extract and products resulting from the fermentation of wheat and soy may be substituted for it (Figure 3) [27,34,35,36]. Moreover, the addition of specific aromatic compounds (e.g., cheese flavour] can be used in order to compensate for the lack of added salt and avoid the flavour changes associated with a significant reduction in salt [37]. In addition, other reformulation strategies must be considered, such as the alteration of food ingredients’ chemical structures, in order to increase sodium availability and release [27].

To support Member States in the reformulation of their products to reduce salt levels, the WHO has also published updated guidelines for the reduction in salt (sodium) concentrations [38,39]. Furthermore, different public health strategies have been aimed at reducing median sodium intake. The Australian government established a salt-reduction initiative for processed meats (bacon, ham/cured meat products, sliced luncheon meat and meat in pastry categories) because these foods contribute 10% of the total daily sodium intake in Australia [40]. After product reformulations, the median sodium intake via processed meats was reduced by 11% in Australia. Similar strategies were established by the Argentinian government in 2011 to reduce salt intake. Unfortunately, the results were uneven. For example, in the Northwest of Argentina, the salt content in street food was monitored while the content in fast foods and artisanal foods was not regulated. As a result, the average salt intake in Argentina still exceeds 12 g/day, in spite of the efforts made [41].

## 3. Reduction in Sugar

In a recent review by the WHO, it was reported that added sugar intake in Europe was greater than 5% of the recommended total energy intake (Figure 2). In fact, children and adolescents are the greatest consumers of sugar-rich foods, such as sugar-sweetened beverages, confections, cakes, pastries and breakfast cereals, with a consumption exceeding the upper limit by 10% [42].

Some epidemiological studies have reported that high carbohydrate intake is associated with a higher risk of total mortality. Thus, in the Prospective Urban Rural Epidemiology (PURE) study, a large epidemiological cohort study of 135,335 individuals aged 35–70 years recruited from 18 countries and followed over a mean of 7.4 years, higher carbohydrate intake (more than 60% of energy) was associated with an increased risk of total mortality (hazard ratio [HR]: 1.28, CI: 1.12–1.46; P_trend_ = 0.001) [43]. In this study, high carbohydrate intake (defined as more than 70% of energy) also showed an increasing risk of major cardiovascular events. It is of note that a systematic review and meta-analysis concluded that high carbohydrate intake, especially from refined sources such as white rice and white bread, is associated with an increased risk of total mortality and cardiovascular events. However, other epidemiological studies, such as the Atherosclerosis Risk in Communities (ARIC) study that included 15,428 adults aged 45–64 years from four US communities, added that a low carbohydrate intake favours the intake of animal-derived protein and fats and also increases overall mortality (HR: 1.20, CI: 1.09–1.32), concluding that a minimal mortality risk is associated with a carbohydrate intake of 50–55%. However, it should be underlined that the increase in mortality observed with low carbohydrate intake was reversed when energy from carbohydrates was replaced by plant-derived proteins or healthy fats [44].

Several meta-analyses that included large prospective studies reported a strong association between the consumption of whole grain and lower total CVD and cancer mortality [45,46,47,48]. Therefore, higher whole grain intake seems to be associated with a lower risk of all-cause mortality. Furthermore, whole grain intake is associated with a lower risk of CVD, type 2 diabetes and cancer [49].

On the other hand, it has been shown that whole grains may modulate appetite, nutrient availability, and energy utilization, thus favouring body weight regulation [50]. Whole grain is richer in dietary fibre, vitamins, minerals, phytoestrogens, phenolic compounds, and phytic acid than refined carbohydrates. Although the processing and reconstituting of whole grain ingredients during food production leads to a reduced capacity of whole grains to regulate body weight [50], doing so results in reductions in BP and cholesterol levels [49]. Polydextrose, resistant starch, inulin, β-glucan and glucomannan are some of the polysaccharides used as ingredients in reformulation. They are good substitutes for starch and sucrose content in foods, and their use reduce the energy density of food products, which may contribute to weight management [51].

In addition, high carbohydrate consumption increases the risk of CVD but mainly diabetes mellitus [52,53], one of the epidemics of the 21st century. Compelling evidence shows that diabetes can be prevented and treated with dietary changes, mainly a reduction in carbohydrate intake [54]. Moreover, high carbohydrate intake increases BP [55], serum triglycerides and small, dense low-density lipoprotein (LDL) cholesterol (the most atherogenic particles), whereas it decreases high-density lipoprotein (HDL) cholesterol [56,57]. Finally, low carbohydrate diets that exchange carbohydrates for a higher intake of protein or fat are commonly used to induce short-term weight loss [58,59], despite conflicting data in relation to their long-term effects on health [60].

Moreover, high sugar intake is strongly associated with a high incidence of these diseases. To limit the increased incidence of diabetes, researchers are currently working to develop synthetic sweeteners as well as the isolation of naturally occurring compounds [61]. Non-caloric synthetic sweeteners, such as saccharin, aspartame, cyclamate, sucralose and acesulfame K, produce undesirable off-tastes when their concentrations increase, shifting from sweet (low concentrations) to bitter or metallic (high concentrations) tastes. Furthermore, their use has been related to psychological problems, mental disorders, bladder cancer, heart failure, brain tumours [61] and, paradoxically, weight gain. The postprandial effects associated with the intake of sweeteners (e.g., saccharin, aspartame, sucralose, acesulfame K) are also associated with weight gain due to the ambiguous psychobiological signals they send out: Sweetness with no calories might lead to overeating and, as a result, loss of control over appetite [61,62]. In the last decades, a large research effort has been put forth to find alternative sugar substitutes, such as novel natural sweeteners (e.g., stevia and monk fruit) and sweet taste enhancers, also called positive allosteric modulators (PAMs). Natural sweeteners have been perceived of as healthful products by consumers, due to the fact that these products are free of synthetic ingredients. However, these natural extracts are unstable and costly to produce due to the difficulties in isolating and growing them outside of their natural environments [61,62]. PAMs are based on the human gene coding for sweet taste receptors and, although they are absolutely tasteless by themselves, potentially enhance sweetness perception [61]. Thus, it has been reported that PAMs are the most effective taste enhancers for reducing energy densities in pastry products in comparison with other reported methods. PAMs enhance sweetness through binding with sweetener receptors without activating them. This binding activity leads to higher affinities of sucrose, fructose and glucose to sweetener receptors [61]. In vitro experiments (cell assays) have reported that SE-2, a PAMs component, enhanced the sweetening effect of sucralose. This effect was not observed in the absence of sucralose [63]. It is important to mention that SE-2 has shown reductions in the range of 4- to 6-fold in the amount of sucralose used while preserving the sweetness intensity [63]. Furthermore, SE-2 could potentially be used to reduce the off-tastes linked to high levels of sucralose present in some products. Other PAMs that are commonly used are SE-3 and SE-4, both of which allow a significant reduction in the total amount of simple sugar (50%) while maintaining the sweetness intensity [63]. Neither SE-2 nor SE-4 produced bitterness, a metallic taste or other undesirable temporal effects (slow onset and/or lingering sweet taste), keeping the taste identical to that of the fully sweetened equivalent [63]. Other PAMs have been studied and used as taste enhancers. Some of these PAM compounds are 2,4 dihydroxybenzoic acid (DHB), 4-amino-5,6-dimethylthieno(2,3-D)pyrimidin-2(1H)-one (ADTP), 3-[(4-amino-2, 2-dioxido-1H-2,1,3-benzothiadiazin-5-yl)oxy]-2,2-dimethyl-N-propylpropanamide (ADBT), 3-Hydroxybenzoic acid (3HB) as well as mixtures of 3HB and DHB, all of which increase the sweetness of sucrose and others common sweeteners [61,64]. Furthermore, 2-methoxy-5-(phenoxymethyl)-phenol and 3′-7-dihydroxy-4′-methoxyflavan have been reported to increase sweetness by 1.3- and 1.6-fold, respectively [61,64].

PAMs have been approved by the Flavour and Extract Manufacturers Association (FEMA) as flavouring ingredients in several foods and beverages [61]. Although the results on this point have been modest, PAMs present a promising path for reducing sugar content in foods and beverages. Nevertheless, safety studies on the use of PAMs in combination with sucrose are required. At present, no safety data have been published by the EFSA or WHO concerning the use of PAMs in the food industry. Moreover, it is necessary to identify more effective and safe PAMs capable of increasing the sweetness intensity up to 20-fold as well as to design PAMs with higher efficacies and specificities [60].

To support Member States in the reformulation of their products to reduce added sugar levels (Figure 3), the WHO has published updated guidelines [60]. Nevertheless, the recommendations designed by the WHO are aligned with the need for some countries to establish dietary programs in order to reduce sugar intake. As an example, in a recent study performed by Garcia et al. [65] the sugar contents in child-oriented breakfast cereals and yoghurts in Latin American countries were compared to the contents in the same products available in UK. The results showed that breakfast cereals from Latin America had higher sugar contents than those from the UK. The authors highlighted that this difference in sugar content might be due to the sugar reformulation policies established in the UK.

## 4. Dietary Fibre

It is known that dietary fibre intake (both types of fibre, i.e., soluble and insoluble) plays a key role in chronic disease prevention, helping to prevent CVD, diabetes mellitus and even in some types of cancer, such as colorectal cancer [66]. In Europe, the daily intake of fibre recommended for adults is 30 g [67], while the intake of fibre is considerably lower (6 g/day) in adults between the ages of 19 to 64 years, according to Scientific Advisory Committee on Nutrition (SANC) survey conducted in the UK [68].

In food reformulation, the inclusion of health-promoting ingredients, such as vitamins and minerals (essential nutrients), phytochemicals, fibre or natural products, is an effective strategy for improving the nutritional characteristics of foods and, consequently, improving diet quality. Nevertheless, up to now, food reformulation concentrated on adding fibre and investigations of its association with health benefits have not been the main aims of many studies [25].

Dietary strategies to achieve and maintain a healthy BMI and avoid failure in weight loss-oriented interventions should concentrate on replacing simple sugars and fats with fibre and protein [69,70]. Fibre-rich foods have lower energy density and high satiety effects, contributing to reduced energy intake or appetite [71]. In particular, β-glucan, lupin kernel fibre, rye bran, whole grain rye, or mixed high-fibre sources have a satiating effect due to their viscosity, which increases retention time in the stomach, and fermentation processes in the colon, which induces satiety responses [71]. Furthermore, fibre could potentially contribute to the energy balance (8 kJ per g fibre), thus increasing energy expenditures from fats and bile acids [72]. Additionally, dietary fibre can exert laxative effects, depending on whether the fibre resists fermentation processes and remains in the large bowel. If so, the result is increased soft and bulky stools [73].

Several observational and interventional studies have showed that fibre-rich diets, such as vegetarian, vegan and Mediterranean diets, are effective in the long-term maintenance of weight loss [74], while some authors have failed to demonstrate their effectiveness [66]. The latter results were most likely not significant due to the heterogeneity in the study designs included in the analyses, the intervention timing or the doses administered.

Energy contributions from simple sugars and starch-rich foods should be reduced by replacing them with dietary fibre from natural or synthetic sources [51]. The main types of fibre used in order to reduce the energy density of carbohydrate-rich foods and produce greater weight loss are polydextrose, resistant starch, inulin, oligofructose, β-glucan and glucomannan [75,76]. Polydextrose is a soluble fibre used widely in the food industry that is fermented partially in the colon, and almost 60% of this fibre is excreted. Furthermore, as an ingredient, it contributes 1 kcal/g in total energy [77]. One of the main characteristics of polydextrose is its ability to induce satiety. Several meta-analyses carried out by Ibarra et al. [76,78] showed that the reformulation of a midmorning snack by adding polydextrose leads to higher satiation and a reduction in the energy intake from the following lunch. This induced satiety seems to follow a dose–response effect. However, up to now, not enough evidence has been produced as to its effect on body weight, and well-designed, long-term clinical trials are required. On the other hand, other fibre ingredients should be considered as well, such as inulin and oligofructose, many of which are soluble and found in many vegetables and fruits in small quantities [79]. They are widely used as ingredients in many reformulated foods and beverages to replace sucrose, or fat content in those products without changing their sensory properties [79,80]. Almost all studies up to now concerning satiety and weight loss have used inulin and oligofructose as supplements rather than as additions to reformulated products. Concurrently, glucomannan, which is a soluble dietary fibre, may produce similar properties, but more studies about its use in reformulated products and weight loss must be performed [81,82]. Finally, β-glucan, which is a soluble fibre present in oats and barley, increases viscosity, attenuates gastric emptying, promotes the production of satiety hormones and acts as an ileal brake mechanism [83]. Oat β-glucan (4 or 8 g/day) intake, used as a solid or liquid matrix, has shown to increase satiety after its intake [84]. However, although recent evidence suggests that oat β-glucan has a positive effect on satiety, its palatability is low [85]. Interestingly, this satiety-inducing effect was not observed after a 3 g of barley β-glucan intake in soup [86].

In summary, food reformulation is a challenge: on the one hand, it is necessary to find the accurate dose of dietary fibre that shows a positive health effect, and, on the other hand, it is necessary to maintain the palatability of the product. The use of fibre as an ingredient in reformulation offers exceptional solubility. Furthermore, it is highly stable, making it ideal for use in low-pH systems and high-temperature processes. Additionally, it provides bulk and desirable mouth feel whilst replacing sugar and fat, and delivers exceptional process stability (low water activity). Moreover, its use has no impact on the final product taste or colour, provides consumer friendly labelling options and provides the possibility of a variety of country-specific claims, including low glycaemic response, prebiotic benefits and increases in calcium absorption, which may help support bone health [87,88,89,90,91,92].

## 5. Reduction in Saturated Fats

Some dietary guidelines have focused on lowering saturated and trans-fat, but not total fat or overall macronutrient composition [93]. Some of these guidelines continue to recommend lowering total fat (<30% of energy from fat), with a corresponding increase in carbohydrate intake [94]. The reduction in total fat, and mainly saturated fat (<10% of total energy intake, Figure 2), is based on the presumption that replacing saturated fatty acids with carbohydrates and unsaturated fats will lower LDL cholesterol levels, thereby reducing CVD events. However, large cohort studies (the Health Professionals Follow-Up and the Nurses’ Health Study) showed an inverse association between total fat intake and total mortality [95]. In fact, these large studies did not find significant associations between saturated fatty acid intake and the risk of CVD when replacement nutrients were not taken into account [96,97,98]. In addition, randomized clinical trials that substituted polyunsaturated fatty acids (PUFA) for saturated fatty acids have also not shown a statistically significant impact on total mortality [98,99]. However, since other evidence has linked high saturated fat intake to the promotion of atherosclerotic vascular disease [100], the current recommendations are to reduce saturated fatty acid intake to less than 10% of total energy [101].

In contrast, several studies have observed an inverse association between monounsaturated fatty acid (MUFA) intake and total mortality. Thus, two large cohort studies, namely, the Health Professional Follow-Up and the Nurses’ Health Study, have shown that lower total mortality is associated with higher MUFA intake [95]. However, in the USA, MUFA comes mainly from meat, whereas the most important source of MUFA in Europe is olive oil. During the follow-up of the 7447 participants included in the PREDIMED trial (Prevención con Dieta Mediterránea), there were 277 cardiovascular events and 323 deaths [8]. Participants in the highest energy-adjusted tertile of baseline total olive oil and extra-virgin olive oil consumption showed reductions of 35% (HR: 0.65; CI: 0.47 to 0.89) and 39% (HR: 0.61; CI: 0.44 to 0.85) in CVD risk, respectively, compared to the reference. Higher baseline total olive oil consumption was associated with a 48% (HR: 0.52; 95% CI: 0.29 to 0.93) reduced risk in cardiovascular mortality. For each 10 g/day increase in extra-virgin olive oil consumption, the risks of CVD and mortality decreased by 10% and 7%, respectively. No significant associations were found for cancer and all-cause mortality [8].

High PUFA intake has been associated with lower total mortality among US men (the Health Professional Follow-Up study) and women (the Nurses’ Health Study) and also in Japanese men [95,102]. Similar results have been observed in a meta-analysis of randomized clinical trials [103], confirming the usefulness of PUFA in preventing CVD and reducing mortality.

Currently, the greatest effort in reformulation is focused on the reduction of saturated fat in milk and milk products, ready-made foods and processed meats [104]. Nevertheless, according to the aforementioned evidence, MUFA and PUFA are two healthy fats that may be used as substitutes for saturated fat or even carbohydrates in the diet and also in the reformulation of bakery products (Figure 3).

Other possible candidates are hydrocolloids, which are long-chain polymers whose main characteristic is forming viscous dispersions and/or gels in contact with water. Among these hydrocolloids, we can cite carboxymethyl cellulose, xanthan gum, high methyl-esterified pectin, low methyl-esterified pectin, sodium alginate and iota-carrageenan. It has been proposed that hydrocolloids replace saturated fat, such as cocoa butter, in chocolate to obtain low-caloric products [105]. Additionally, the result from the enzymatic hydrolysis of oat flour rich in soluble fibre (β-glucan) can be used as substitute for fat, maintaining physical and rheological properties in formulations with 40% to 60% substitutions [105]. In addition, the monoglyceride organogels (monoglyceride emulsion and sunflower or monoglyceride and palm oil) are considered as alternatives for saturated fats and trans-fats [105]. In fact, it has been reported the substitution of hydrogel with sunflower oil for palm oil might lead to significant reductions in saturated fat in bread (81% w/w) [106]. Nevertheless, up to now, developing alternative products to replace fat remains a challenge, as fat has several important physicochemical and sensory attributes that are very difficult to substitute for, such as a smooth, creamy, rich texture; a milky, creamy appearance; desirable flavour and its satiating effect.

Other ingredients or compounds with structural and physicochemical properties similar to those of triglycerides are olestra, which is obtained via the trans- or inter-esterification of sucrose (mixture of several esters of sucrose plus the addition of six to eight long-chain fatty acids), caprenin (caprocaprylobehenic added to a triacylglyceride) and salatrim (short and long acyl triaclyglyceride molecules), all of which are used widely as fat substitutes. These ingredients are characterized as indigestible by humans and reduce the caloric contents of products more than triglycerides do. However, their intake may make the absorption of dietary lipophilic vitamins, such as vitamins A, D, E and K, difficult [51].

## 6. Reduction in Trans-Fats

Trans-fats contribute about 1.2% of total energy intake in Western diets [100] and are produced industrially through the partial hydrogenation of plant oils in the presence of a metal catalyst, high heat or a vacuum. They can also occur naturally in meat and dairy products through the biohydrogenation of unsaturated fatty acids via bacterial enzymes in ruminant animals. These products share the characteristic of having at least one double bond in the “trans” rather than the “cis” configuration. The majority of trans-fats present in diets are of industrial origin [20].

Several cohort studies have reported that trans-fatty acids are associated with increased all-cause mortality, sudden death and fatal colon and breast cancer [93,94]. In addition, trans-fatty acid intake is also related to a higher risk of coronary heart disease via blood lipids and pro-inflammatory processes [95]. A 2% increase in energy intake from trans-fats is associated with a 25% increased risk of coronary heart disease and a 31% increase in cardiovascular mortality [93]. All these effects have been observed with industrially produced trans-fats. No associations were observed for ruminant-produced trans-fats.

In addition, trans-fat intake seems to be associated with the development of obesity, diabetes, neurodegenerative disease (Alzheimer’s), some types of cancer (mainly breast cancer), impaired fertility, endometriosis and cholelithiasis [107].

One of major improvements achieved to date is the elimination of trans-fats in margarine, pastries, cakes and biscuits in Europe [108]. This elimination was made possible through both legislative and voluntary measures. In addition, manufacturers have begun to replace saturated fats with MUFAs and PUFAs (Figure 3) [109]. The substitution of other vegetable oils, such as olive oil, which are characterized by no or low trans-fats and high cis-unsaturated fatty acids, for hydrogenated vegetable oils would improve the nutritional quality of food and add the health benefits associated with lower intakes of trans-fats [51]. In this sense, oil seed producers and agriculturists must increase the supply of substitute oils [51].

Therefore, trans-fats in processed food should be kept as low as possible and should always make up <1% of energy, primarily to reduce the risk of ischemic heart disease and stroke (Figure 2).

## 7. Comments

The large increase in obesity over the past 30 years has been attributed to several potential sources, including increases in caloric intake, changes in diet composition, declining levels of physical activity and changes in other factors, such as the gut microbiome. Nowadays, the recommendations for the general population are directed at upgrading the nutritional quality of overall dietary patterns, i.e., increasing the adherence to a healthy diet such as the MD. In addition, the restriction of unhealthy foods, such as processed foods, is also highlighted. Since most people like/consume bakery and pastry products, the reformulation of these processed foods is an interesting approach for reducing excessive energy, salt, saturated fat and sugar intakes. Food industry advances have provided a wide range of components that can be used as substitutes for these nutritional objectives, such as aromatic compounds for reducing salt content, polysaccharides, fat substitutes, olive oils rich in MUFA and PUFA and PAMs. Reformulation is necessary to improve the nutritional quality of products and, consequently, reduce the associated risk of developing chronic diseases, such as CVD, diabetes and cancer.

## Figures and Tables

**Figure 1 nutrients-12-01709-f001:**
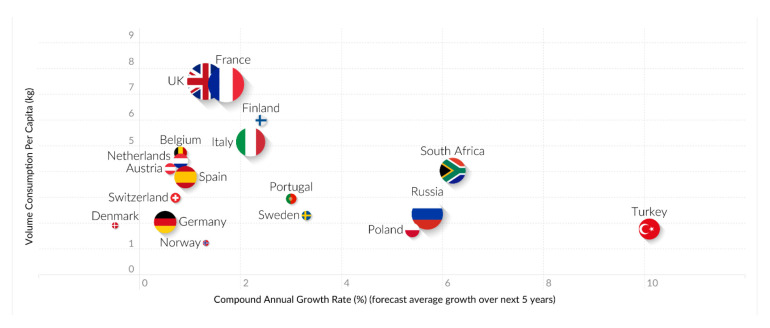
Trends at the EMEA level in Europe, the Middle East and Africa. Source: Mintel, A Year of Innovation in Cakes and Sweet bakery, 2019.

**Figure 2 nutrients-12-01709-f002:**
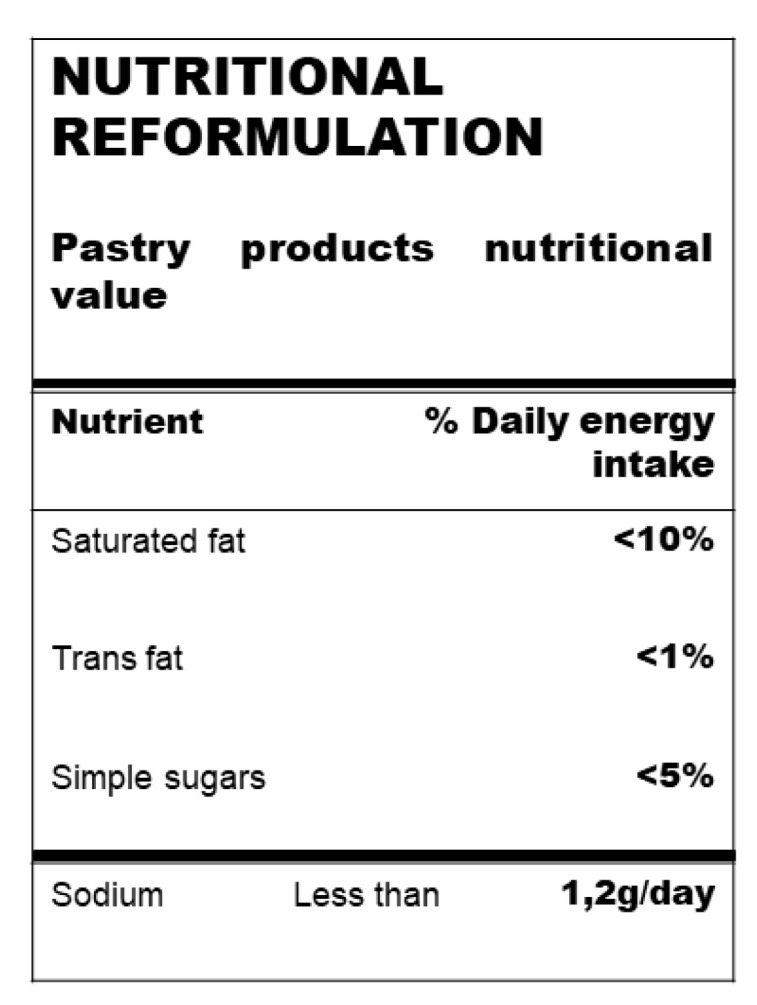
Recommended intake levels expressed as % of daily energy intake (kcal/day) of saturated and trans-fat and simple sugars and grams of sodium *. * Useful in pastry and bakery product reformulation.

**Figure 3 nutrients-12-01709-f003:**
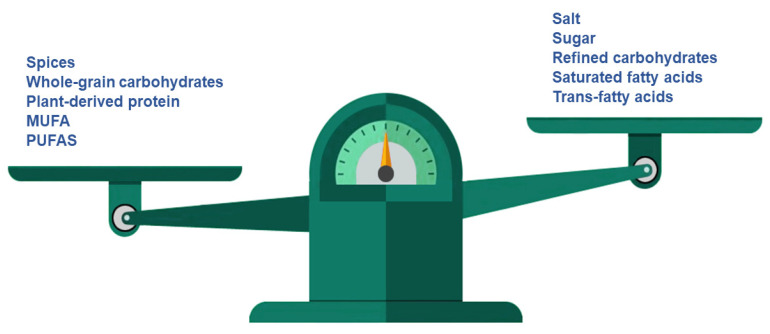
Reformulation proposal for healthier bakery and pastry products *. * Potassium chloride, monosodium glutamate, nucleotids, polysaccharides, natural sweeteners, positive allosteric modulators (PAMs), caprenin, salatrin and olestra may be used to enhance flavour.

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
