# Peer review of "Reformulation of Pastry Products to Improve Effects on Health"

_nutrients, 2020, doi:10.3390/nu12061709_

Round 1
Reviewer 1 Report
Overall, the topic of such review is of relevance for the scientific community. However, the manuscript in its current form appears rather preliminary and not really carefully crafted, resembling more a "draft" than a final version.
Much more attention, in terms of text organization, accuracy of definitions, and quality/consistency of figures should be paid during the revision process, in order to provide a more polished manuscript.
The following papers should be cited:
Sodium Levels of Processed Meat in Australia: Supermarket Survey Data from 2010 to 2017.
Nutrients. 2018 Nov 5;10(11). pii: E1686. doi: 10.3390/nu10111686.
We are What We Eat: Impact of Food from Short Supply Chain on Metabolic Syndrome.
J Clin Med. 2019 Nov 23;8(12).
Sugar Content and Nutritional Quality of Child Orientated Ready to Eat Cereals and Yoghurts in the UK and Latin America; Does Food Policy Matter?
Nutrients. 2020 Mar 23;12(3).
Sodium Content in Commonly Consumed Foods and Its Contribution to the Daily Intake.
Nutrients. 2019 Dec 21;12(1). pii: E34. doi: 10.3390/nu12010034.
Some references should be updated.
Author Response
nutrients- 805363
REVIEWER #1:
- Overall, the topic of such review is of relevance for the scientific community. However, the manuscript in its current form appears rather preliminary and not really carefully crafted, resembling more a "draft" than a final version.
Much more attention, in terms of text organization, accuracy of definitions, and quality/consistency of figures should be paid during the revision process, in order to provide a more polished manuscript.
Comment 1: Thank you for your comment. We have reviewed all the manuscript. A new section has been created (Dietary fiber). The changes are highlighted in yellow (lines 252-320).
Besides, according to Reviewer#2, we have deleted the paragraph about Mediterranean diet. The changes are highlighted in yellow (lines 46-67).
Finally, we have included references indicated by reviewer#1. The changes are highlighted in yellow (Reference section, references 18, 40, 41 and 65).
The following papers should be cited:
Sodium Levels of Processed Meat in Australia: Supermarket Survey Data from 2010 to 2017. Nutrients. 2018 Nov 5;10(11). pii: E1686. doi: 10.3390/nu10111686.
We are What We Eat: Impact of Food from Short Supply Chain on Metabolic Syndrome. J Clin Med. 2019 Nov 23;8(12).
Sugar Content and Nutritional Quality of Child Orientated Ready to Eat Cereals and Yoghurts in the UK and Latin America; Does Food Policy Matter? Nutrients. 2020 Mar 23;12(3).
Sodium Content in Commonly Consumed Foods and Its Contribution to the Daily Intake. Nutrients. 2019 Dec 21;12(1). pii: E34. doi: 10.3390/nu12010034.
Comment 3: Thank you for Reviewer’s comment. We have included all references. The changes are highlighted in yellow (Reference section, references 18, 40, 41 and 65).
- Some references should be updated.
Comment 3: Thank you for your comment. We have updated references.

Reviewer 2 Report
This manuscript reports many interesting tips to improve the nutritional quality of pastry.
The paper may be of interest and fits with the purpose of the Journal. However, some revisions are needed.
General comment
The reduction of saturated and trans fatty acids, salt and sugar are pivotal modifications to obtain healthier pastry products. However, adding fibre could be another relevant issue to address.
Please add a paragraph about fibre.
Specific comments
The strategies reported by the authors (low saturated and trans fatty acids, salt and sugar) are common features of healthy dietary pattern. In addition, the reduction of this dietary components are healthier per se.
Therefore I think the authors should not mention Mediterranean diet in the introduction (lines 47 and over).
Moreover, pastry is not a key feature of the Mediterranean diet.
Author Response
nutrients- 805363
REVIEWER #2:
- This manuscript reports many interesting tips to improve the nutritional quality of pastry. The paper may be of interest and fits with the purpose of the Journal. However, some revisions are needed.
General comment
The reduction of saturated and trans fatty acids, salt and sugar are pivotal modifications to obtain healthier pastry products. However, adding fibre could be another relevant issue to address.
Please add a paragraph about fibre.
Comment 1: Thank you for your comment. We have reviewed all the manuscript. A new section has been created (Dietary fiber). The changes are highlighted in yellow (lines 252-320).
- Specific comments
The strategies reported by the authors (low saturated and trans fatty acids, salt and sugar) are common features of healthy dietary pattern. In addition, the reduction of this dietary components is healthier per se.
Therefore, I think the authors should not mention Mediterranean diet in the introduction (lines 47 and over).
Moreover, pastry is not a key feature of the Mediterranean diet.
Comment 2: Thank you for your comment. We agree with Reviewer#2, thereby we have deleted the paragraph about Mediterranean diet. The changes are highlighted in yellow (lines 46-67).

Round 2
Reviewer 1 Report
-
Author Response
1. Moderate English changes required
Comment 1: Our manuscript has been checked by a professional English editing service and we attach you the certificate.

Reviewer 2 Report
The manuscript has improved after the revision. I think is suitable for publication in the present form.
Author Response
Thank you to Reviewers for your suggestions